# Specific hippocampal representations are linked to generalized cortical representations in memory

Jai Y. Yu[1], Daniel F. Liu[1,2], Adrianna Loback[3], Irene Grossrubatscher[2] & Loren M. Frank[1,4,5]

Memories link information about specific experiences to more general knowledge that is abstracted from and contextualizes those experiences. Hippocampal-cortical activity patterns representing features of past experience are reinstated during awake memory reactivation events, but whether representations of both specific and general features of experience are simultaneously reinstated remains unknown. We examined hippocampal and prefrontal cortical firing patterns during memory reactivation in rats performing a well-learned foraging task with multiple spatial paths. We found that specific hippocampal place representations are preferentially reactivated with the subset of prefrontal cortical task representations that generalize across different paths. Our results suggest that hippocampal-cortical networks maintain links between stored representations for specific and general features of experience, which could support abstraction and task guidance in mammals.

[1] UCSF Center for Integrative Neuroscience and Department of Physiology, University of California San Francisco, San Francisco, CA 94143, USA. [2] University of California Berkeley, Berkeley, CA 94720, USA. [3] Princeton University, Princeton, NJ 08544, USA. [4] Howard Hughes Medical Institute, University of California, San Francisco, CA 94143, USA. [5] Kavli Institute for Fundamental Neuroscience, University of California, San Francisco, CA 94143, USA. Correspondence and requests for materials should be addressed to L.M.F. (email: loren@phy.ucsf.edu)

How does the brain represent the content of individual experiences with respect to their more general significance? Take, for example, memories of using staircases in your apartment building, home, or workplace. You can most likely recall unique memories of walking up a specific staircase connecting two particular floors. These individual memories do not exist in isolation but are connected to more general knowledge, such as staircases are used for traveling between floors. It is not known how activity patterns in the brain support the link between specific and general features of experience, which is necessary to correctly embed individual memories in broader knowledge structures.

The hippocampus and prefrontal cortex (PFC) are thought to play complementary roles in maintaining these types of

knowledge[1–6]. The hippocampus is important for memories of specific experiences[7–9], while the PFC supports memories that generalize across experiences[10–12]. In the context of rodent spatial exploration, individual experiences engage hippocampal place cell ensembles that represent specific locations[13–15] and spatial trajectories[16]. In contrast, PFC firing patterns seen during behavior are highly heterogeneous and are more often related to different behavioral stages or the structure of an ongoing task[17–29] than locations in space[18–20].

Stored associations between hippocampal and PFC representations are transiently reinstated at the time of hippocampal sharp-wave ripple (SWR) events[30–33]. It remains unknown whether these associations can link specific hippocampal representations of locations to more general PFC representations

**Fig. 1** Example CA1 and PFC task activity patterns. **a** Occupancy normalized spatial firing maps for 2 CA1 and 4 PFC cells that were simultaneously recorded. The maximum firing rate for each cell is indicated below each panel. **b** Time normalized trial firing rate maps for the cells in **a**. The vertical dotted line separates well and path trial phases. Horizontal solid lines separate trials on each trajectory. Five example trials are shown for each trajectory. The trajectory for each group of trials is indicated by the schematic to the left of column 1. Solid and dotted arrows indicate the two directions of travel on a path. The firing rate color scale is the same as in **a**–**c**. Median firing rate for each trajectory. Line color corresponds to arrow color scheme in **b**–**d**. Distribution of pairwise Pearson's correlation for trial firing profile across all trials. All-trial similarity ($R_{median}$) is the median of the distribution and is indicated by the arrowhead. **e** Distribution of pairwise Pearson's correlation for firing profile of trials on the trajectory with the highest median pairwise correlation. Maximum within-trajectory similarity ($R_{max}$) is the median of the distribution and is indicated by the arrowhead

related to task structure. Previous findings established that early during learning, the degree of hippocampal-PFC co-firing during ongoing experience predicts their coactivity during SWRs[30,32]. As a result, activity during SWRs reflects spatial associations related to where in the environment a particular set of hippocampal and PFC cells are co-active. However, once the environment and task are familiar, hippocampal-PFC coactivity during ongoing experience is no longer predictive of SWR coactivity[32]. Nonetheless, many PFC cells continue to show modulation during SWRs[32–34], which suggests that hippocampal-PFC reactivation patterns in familiar settings could reflect associations between spatially specific hippocampal representations and other, potentially more general task-related, PFC representations.

To determine the structure of stored associations between hippocampal and PFC representations, we asked which PFC representations are concurrently reactivated with hippocampal representations in a familiar task. A null-hypothesis predicts that an unbiased selection from the pool of heterogeneous PFC representations remains linked with hippocampal representations. However, we found a select subset of PFC representations for general features shared across individual experience are enriched, which provides a potential mechanism for generalization across individual experiences.

## Results

**Contrasting CA1 and PFC representations during ongoing experience.** We examined hippocampal-cortical representations by recording activity simultaneously in the hippocampus and PFC of well-trained rats performing a spatial foraging task in which they are required to travel across multiple paths to obtain food reward. We make the distinction between paths, which physically connect reward locations, and trajectories, which are the combinations of paths used by the rat to travel between reward locations. Given the contrasting representations in the hippocampus and PFC, which potentially encode different features of experience, we designed the task to examine representations for "specific" and "general" features of experience in a spatial context. We refer to "specific" representations as those expressed on individual trajectories and "general" as representations expressed across multiple trajectories that capture their common features. The task had four potential reward locations (wells) interconnected by paths. At any given time, only two wells could dispense reward. To receive reward the animal needed to find these two wells and visit them in alternation (Supplementary Fig. 1). By switching the rewarded wells within and/or between sessions or days, we encouraged the animal to travel between

wells via different trajectories that could consist of any combination of paths. We defined a trial as the time between consecutive well location visits. To compare hippocampal (CA1) and PFC activity on different trajectories, we normalized CA1 and PFC activity by dividing each trial into 36 time bins (see Methods). We also confirmed the animal displayed similar movement patterns on different trials and trajectories (Supplementary Fig. 2).

As expected, CA1 cells ($N = 234$) showed location specific activity (Fig. 1a–c, CA1 cells 1–2). PFC cells ($N = 578$) showed diverse activity patterns including those that varied across different trajectories (Fig. 1a–c, PFC cells 1–2) and those that were active at similar trial phases across different trajectories, consistent with representations of task structure (Fig. 1a–c, PFC cells 3–4). We described the firing pattern of each cell using two parameters: all-trial similarity and maximum within-trajectory similarity. All-trial similarity captures the consistency of firing profiles between all pairs of trials and is thus a measure of generalization. All-trial similarity was quantified as the median of pairwise Pearson's correlations between firing profiles across all pairs of trials ($R_{median}$, Fig. 1d). We found all-trial similarity also captured firing similarity across different trajectories as shown by its strong correlation with intertrajectory-trial similarity (Intertraj. $R_{median}$), which is calculated from pairs of trials on different trajectories (Supplementary Fig. 3). In contrast, maximum within-trajectory similarity captures the reliability of the cell's activity in representing aspects of a specific spatial trajectory. We quantified maximum within-trajectory similarity by calculating a similarity score separately for trials on each trajectory and then taking the maximum of those scores across trajectories ($R_{max}$, Fig. 1e).

Across the population, CA1 cells typically had low $R_{median}$ but high $R_{max}$ values (Fig. 2a–c, cyan), reflecting their spatially specific and reliable location representations. The low $R_{median}$ values also suggest that in our task, the representation of specific spatial location dominates other forms of task representations found in the hippocampus, including context[35] and time[36]. In contrast, PFC cells had a wide range of $R_{median}$ and $R_{max}$ values (Fig. 2a–c, orange), reflecting the heterogeneity of representations that is typical for this brain region. Nonetheless, a subset of these PFC cells had similar (high $R_{median}$) and reliable (high $R_{max}$) activity patterns even between trials on trajectories with different lengths (e.g., Fig. 1 PFC cell 4 and Supplementary Fig. 4).

We next asked how PFC activity relates to the task, and whether the population as a whole provided a representation of the various trial phases. As a population, PFC cells were active across the entire range of trial phases (Fig. 3a–d) and individual

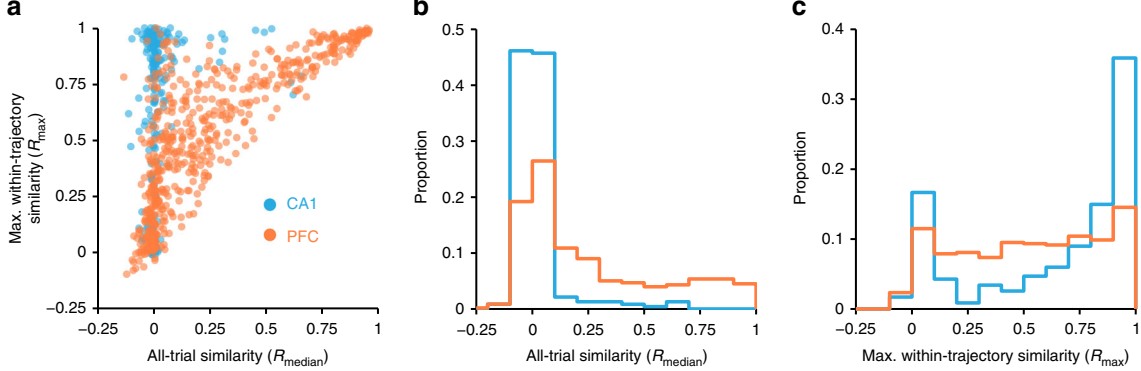

**Fig. 2** Location specific CA1 and diverse task PFC activity patterns. **a** Scatter of $R_{max}$ and $R_{median}$ for CA1 (cyan, $N = 234$) PFC cells (orange, $N = 556$). **b** Distribution of $R_{median}$ (from **a**) for CA1 (cyan, $N = 234$) and PFC cells (orange, $N = 578$). Kolmogorov–Smirnov test: ***$p < 10^{-4}$. **c** Distribution of $R_{max}$ (from **a**) for CA1 (cyan, $N = 234$) and PFC cells (orange, $N = 556$). Kolmogorov–Smirnov test: ***$p < 10^{-4}$

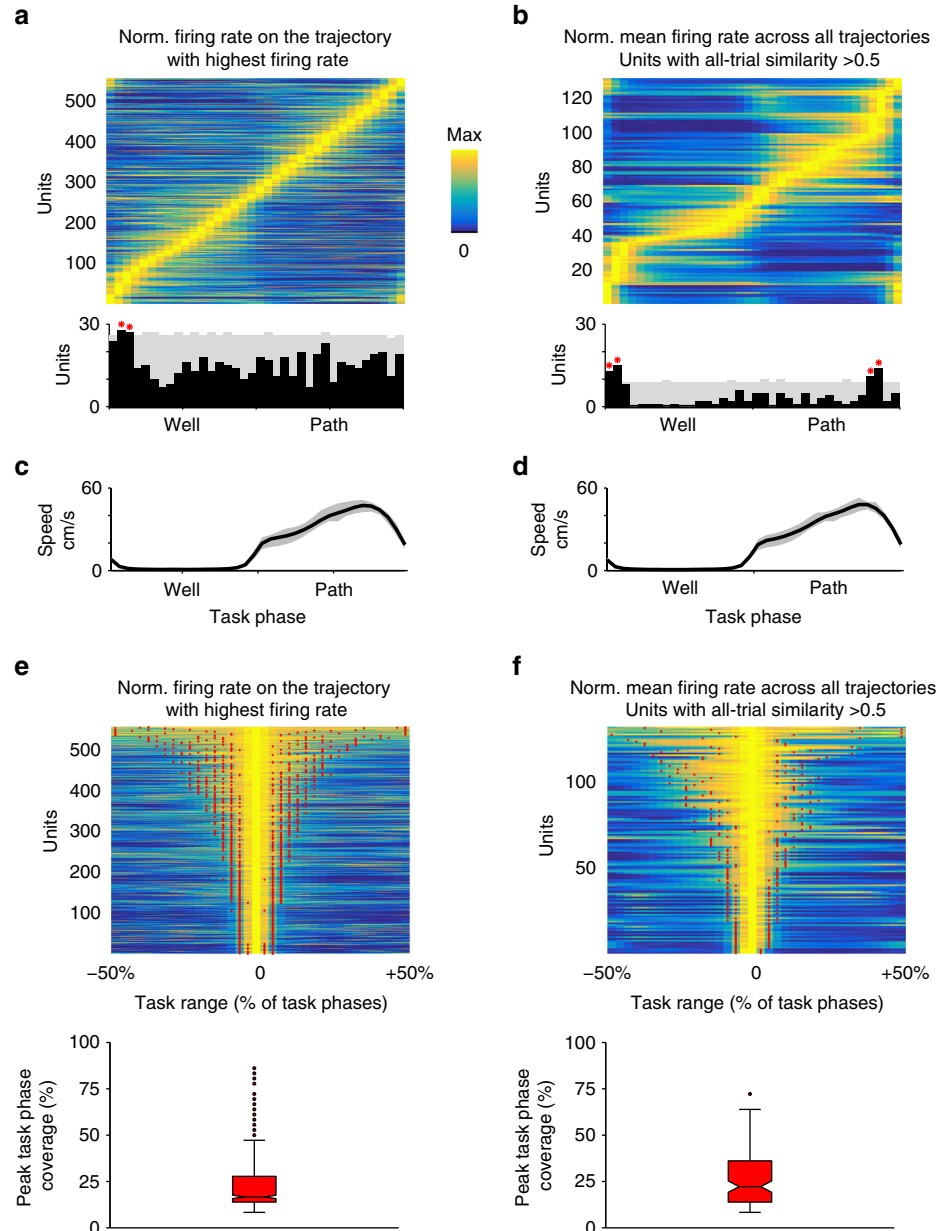

**Fig. 3** PFC activity is distributed across and shows specificity for task phases. **a** Upper panel shows normalized mean firing rate for each cell on the trajectory with the highest mean firing rate. Cells ($N = 556$) are sorted by the task phase with the maximum firing rate. PFC firing spans the entire trial phase range. Lower panel shows the distribution of peak firing task phase (black bars) and the 99% confidence interval of expected distribution from 5000 permutations (gray). Red asterisks mark task phases (near well entry) with a greater than expected numbers of cells. **b** Upper panel shows normalized mean firing rate for each cell across all trajectories. Only cells with similar firing across different trajectories ($R_{median} > 0.5$) are shown. Cells ($N = 131$) are sorted by the task phase with the maximum firing rate. Lower panel shows the distribution of peak firing task phase (black bars) and the 99% confidence interval of expected distribution from 5000 permutations (gray). Red asterisks mark task phases (near well entry) with a greater than expected numbers of cells. **c, d** Corresponding median speed profile for **a** and **b** respectively. The interquartile range is in gray. **e** Normalized mean firing rate for each cell on the trajectory with the highest mean firing rate (same as **a**) aligned to the trial phase where each cell's rate was maximum and sorted by peak task phase coverage (bounds marked by red dots). Bottom panel shows the corresponding box plot of peak task phase coverage. Most PFC cells are active over less than half of task phases. **f** Normalized mean firing rate for each cell across all trajectories aligned to the trial phase with maximum firing rate (same as **b**) and sorted by peak task phase coverage. Only cells with similar firing across different trajectories ($R_{median} > 0.5$) are shown. Peak task phase coverage bounds indicated by red dots. Bottom panel shows the corresponding box plot of peak task phase coverage. Peak task phase coverage is the number of contiguous task phases around the peak with firing rate exceeding 66.67% of maximum

cells showed peak firing that was restricted to certain trial phases (Fig. 3e–f). We also found a significant overrepresentation in the population for trial phases around well location entry (Fig. 3a–b), which could correspond to salient task elements such as the choice point before the reward location and receiving reward. We

also note that these representations were unlikely to be solely explained by an animal's movement speed (Supplementary Fig. 5). Instead, firing properties of these cells are consistent with representations for stages of a trial, and cells with similar firing patterns across trajectories could therefore encode general task

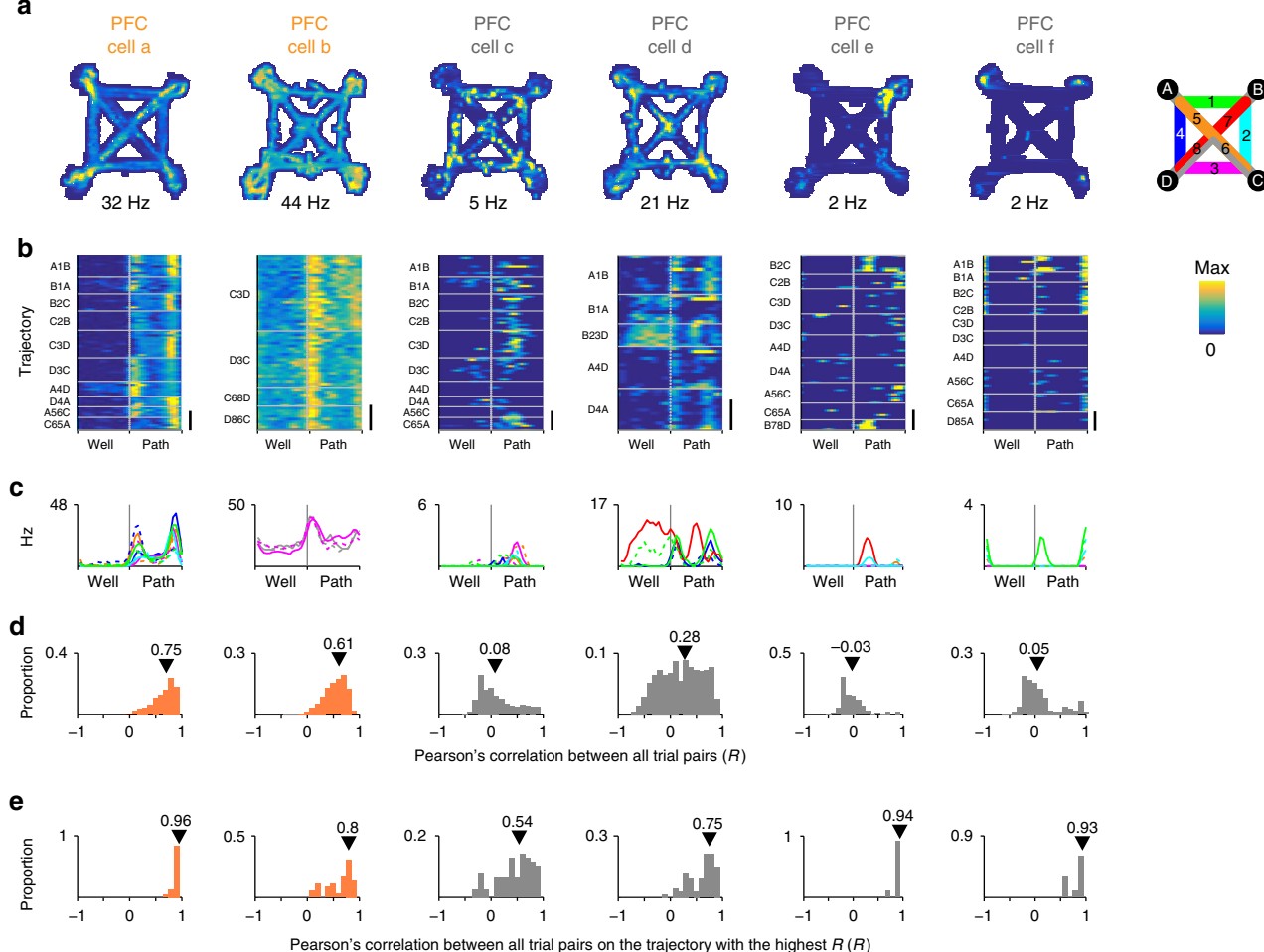

**Fig. 4** Task firing patterns for example SWR reactivated and not reactivated path-preferring PFC cells. **a** Occupancy normalized spatial firing maps for path-preferring PFC cells. Maximum firing rate of each cell is indicated below each panel. Cells a and b show significant SWR reactivation. Cells c–f are not reactivated. Cells a and c were recorded simultaneously. **b** Time normalized trial firing rate map for the cells in **a**. The vertical dotted line separates well and path trial phases. Horizontal solid lines separate trials on each trajectory. Each trajectory is labeled to indicate the start and the end well locations (letters), and paths taken (number) according to the maze schematic in **a**. Maximum firing rate and color scale are the same as in **a**. Scale bar = 10 trials. **c** Median firing rate for each trajectory. Line color key is indicated by maze schematic in **a–d**. Distribution of pairwise Pearson's correlation for trial firing profile across all trials. All-trial similarity ($R_{median}$) is the median of the distribution and is indicated by the arrowhead. **e** Distribution of pairwise Pearson's correlation of trials on the trajectory with the highest median pairwise correlation. Maximum within-trajectory similarity ($R_{max}$) is the median of the distribution and is indicated by the arrowhead

stage information shared between trials with different spatial locations and geometries.

**Generalized PFC representations are reactivated preferentially**. We next asked whether a specific subset of these heterogeneous task-related PFC representations remains linked with hippocampal spatial representations during awake SWR memory reactivation. Since hippocampal trajectory representations are frequently reactivated during SWRs[37–39], our goal was to understand how the reactivation of different trajectory-associated representations are coordinated between CA1 and PFC once the task became familiar. We restricted our analyses to SWRs that occurred at the reward wells (see Methods section) and focused on hippocampal-cortical reactivation in well-trained animals with at least 5 days of exposure to the task.

We first selected CA1 and PFC cells that were preferentially active on paths[33]. We then examined SWR events that reactivated these path-active CA1 cells and asked whether during these SWRs, concurrently reactivated PFC cells (i.e., cells with firing

rate increases) ($N = 35$ cells) had task activity patterns that differed from the selected PFC cells that were not reactivated (i.e., cells without firing rate change) ($N = 161$ cells) (Supplementary Fig. 6). We restricted our analyses to PFC and CA1 path-preferring cells and their respective SWRs to ensure we can adequately sample specific-to-general mapping between multiple hippocampal and PFC representations, since we rarely had days with more than one CA1 well place cell recorded simultaneously with PFC cells.

We found that the population of reactivated PFC cells were substantially enriched for cells with high all-trial similarity ($R_{median}$), which by definition also had high maximum within-trajectory similarity ($R_{max}$) (Fig. 4a–e PFC cells a–b and Fig. 5a–c orange). By contrast, PFC cells that did not participate in SWR reactivations tended to show activity that varied from trial to trial or between trajectories. The activity pattern of this population was mostly dissimilar across trajectories ($R_{median}$ distribution dominated by low values) and varied in within-trajectory similarity ($R_{max}$ values distributed uniformly) (Fig. 4a–e PFC cells c–f and Fig. 5a–c gray).

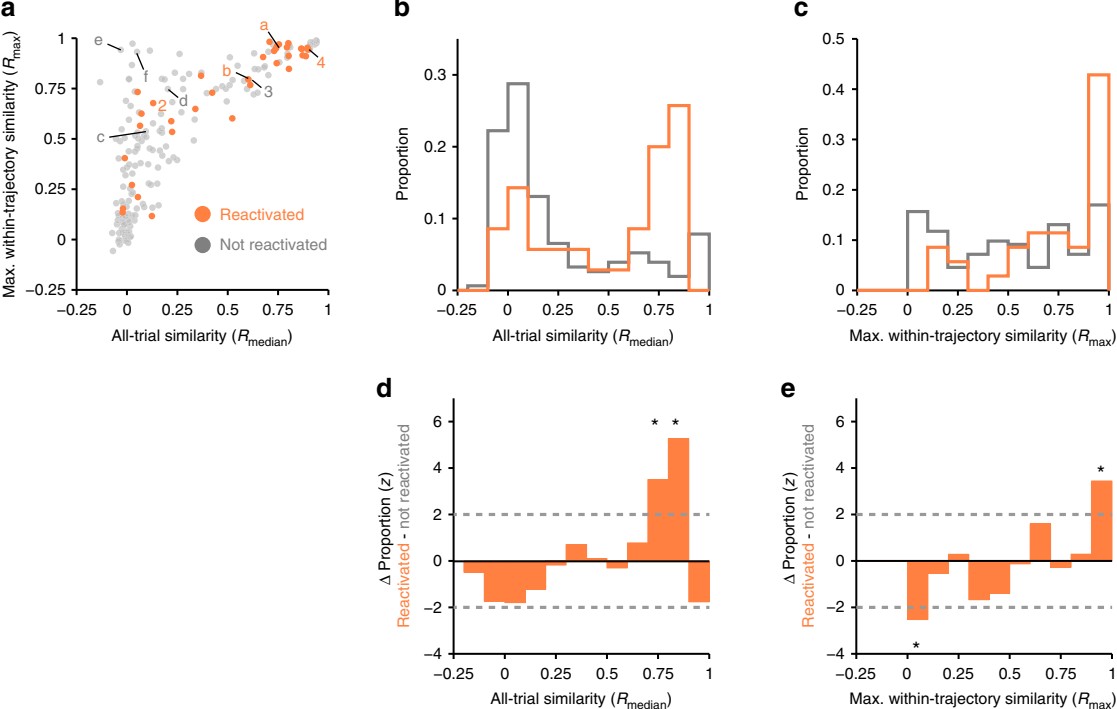

**Fig. 5** SWR reactivated path-preferring PFC cells show higher activity similarity across different trajectories. **a** Scatter of $R_{max}$ and $R_{median}$ for SWR reactivated (orange, $N = 35$) and non-reactivated ($N = 161$, gray) PFC cells. Cells from Figs. 1 (cells 2 and 4) and Fig. 4a (cells a-f) are labeled. **b** Distribution of $R_{median}$ (from **a**). Kolmogorov–Smirnov test: ***$p < 10^{-3}$. **c** Distribution of $R_{max}$ (from **a**). Kolmogorov–Smirnov test: ***$p < 10^{-3}$. **d** Difference in the $R_{median}$ distributions between path SWR reactivated and not reactivated PFC cells normalized using a permutation test (see Methods section). **e** Difference in the $R_{max}$ distributions between path SWR reactivated and not reactivated PFC cells normalized using a permutation test (see Methods section). Dotted lines indicate ±2 S.D.

The differences between reactivated and non-reactivated PFC cells were clear when we examined the distribution of $R_{median}$ and $R_{max}$ relative to that of the entire path-preferring PFC population. If reactivated PFC cells were an unbiased subsample from the population of path-preferring PFC cells, we would have expected to find comparable $R_{median}$ and $R_{max}$ distributions in both the reactivated and non-reactivated populations. Instead, we found a significantly larger fraction of PFC cells with high $R_{max}$ and $R_{median}$ in the SWR reactivated population than expected given the baseline population distribution (Fig. 5d–e). We further reproduced these findings using intertrajectory-trial similarity, which confirms reactivated cells show higher similarity in firing across different trajectories (Supplementary Fig. 7). The difference in SWR engagement could not be explained by differences in peri-SWR firing rate that might have influenced our ability to detect positive SWR modulation in PFC cells (Supplementary Fig. 8). We verified that our results are not biased by potentially including cells that were recorded across multiple days (Supplementary Figs. 9 and 10). Our results also remained consistent when all SWRs were included (Supplementary Fig. 11), likely because the majority of SWRs contain path location reactivations[33].

Importantly, we found that trial similarity measures can better account for the participation of PFC cells in SWR reactivation than other firing correlates, including co-activity with CA1 cells during behavior. We constructed Generalized Linear Models (GLMs) to assess the contribution of different firing properties to predicting whether a PFC cell is modulated during SWRs. These included trial similarly as well as mean trial firing rate, activity coverage on a trial, speed-firing rate$_{residual correlations}$ and proximity of peak firing to wells, defined as follows: activity coverage on a trial is the proportion of consecutive trial phase

bins with firing rate exceeding 2/3 of the peak rate; speed-firing rate$_{residual correlation}$ captures the correlation between firing rate and speed while controlling for effect of trial phase (see Methods section); peak firing proximity to wells is the minimum number of trial bins from the peak firing rate bin to a well bin. This final measure was designed to control for any effects of proximity of PFC spiking activity during behavior to activity during SWRs. We found that a model including all of these firing rate related variables explained only 3.8% of the prediction variance of PFC SWR participation. In contrast, models with the addition of only one of the trial similarity measures, Intratraj. $R_{median}$, $R_{max}$ or Intertraj. $R_{median}$, captured between 7.4 and 10.8% of the variance (Supplementary Fig. 12 red versus orange bars), which represent a 1.95–2.84 fold improvement. The results of the GLMs demonstrate trial similarity measures were better predictors than the other variables (Table 1 and Supplementary Fig. 12).

Similarly, we confirmed that in a familiar task setting, pairwise CA1-PFC coactivity during ongoing experience was a poor predictor of their reactivation[32]. Peak coactivity during ongoing experience within an 100 ms window was only weakly related to their coactivity during reactivation (Supplementary Fig. 13A). We also found coactivity within the duration of a typical trial (~5 s) showed a weak negative correlation with coactivity during reactivation (Supplementary Fig. 13B). This reflects the existence of many cell pairs that were strongly coactive during ongoing experience but were nevertheless not reactivated together. In contrast, all-trial similarity of individual PFC cells was a much stronger predictor of reactivation (Supplementary Fig. 13C–E). Thus, while it remains possible that there are other aspects of PFC activity that also predict SWR reactivation, our findings point to firing similarly across

**Table 1 Firing similarity measures predict participation in SWR reactivation**

| GLM | $X_1$ | $\beta_1$ standard errors | p | $X_2$ | $\beta_2$ standard errors | p |
|---|---|---|---|---|---|---|
| 1 | All-trial $R_{median}$ | 3.61 | 0.0003 | **Firing rate** trial mean | −0.17 | 0.86 |
| 2 | All-trial $R_{median}$ | 2.80 | 0.0052 | **Firing rate** trial max | −0.04 | 0.97 |
| 3 | Intertraj. $R_{median}$ | 3.36 | 0.0008 | **Firing rate** trial mean | −0.04 | 0.97 |
| 4 | Intertraj. $R_{median}$ | 2.52 | 0.0118 | **Firing rate** trial max | 0.19 | 0.85 |
| 5 | $R_{max}$ | 3.19 | 0.0014 | **Firing rate** trial mean | 0.55 | 0.58 |
| 6 | $R_{max}$ | 2.52 | 0.0117 | **Firing rate** trial max | 1.00 | 0.32 |
| 7 | All-trial $R_{median}$ | 4.09 | 0.00004 | **Trial coverage** Most active trajectory | 0.59 | 0.55 |
| 8 | All-trial $R_{median}$ | 3.86 | 0.0001 | **Trial coverage** All trials | 0.55 | 0.58 |
| 9 | Intertraj. $R_{median}$ | 3.57 | 0.0004 | **Trial coverage** Most active trajectory | 0.70 | 0.48 |
| 10 | Intertraj. $R_{median}$ | 4.15 | 0.00003 | **Trial coverage** All trials | 0.92 | 0.36 |
| 11 | $R_{max}$ | 3.93 | 0.0001 | **Trial coverage** Most active trajectory | 0.89 | 0.38 |
| 12 | $R_{max}$ | 3.59 | 0.0003 | **Trial coverage** All trials | 0.93 | 0.35 |
| 13 | All-trial $R_{median}$ | 3.48 | 0.0005 | **Speed-Firing rate** residual correlation | −0.68 | 0.50 |
| 14 | Intertraj. $R_{median}$ | 3.20 | 0.0014 | **Speed-Firing rate** residual correlation | −0.62 | 0.53 |
| 15 | $R_{max}$ | 2.99 | 0.0028 | **Speed-Firing rate** residual correlation | 0.33 | 0.74 |

A Generalized Linear Model (GLM) with two predictors and a logistic link function was used to predict SWR reactivation based on firing similarity and firing measures. Firing similarity measures ($X_1$) but no other firing measures ($X_2$) significantly predicted SWR reactivation as indicated by the significance of the $\beta$ values. SWR reactivation was modeled as a binomial distribution (0 for non-reactivated and 1 for reactivated). $\beta$ values are expressed as standard errors to allow comparison between predictors. Firing rate measures are mean and peak rates of mean trial firing across all trajectories. Trial coverage is defined as the number of consecutive trial phase bins around the peak trial phase with firing rate exceeding 2/3 of maximum rate. Trial coverage was calculated for the mean trial firing rate for trials on the most active firing trajectory or for all trajectories. Speed-firing rate correlation is the correlation for speed and firing rate residuals for each trial phase for trials on the same trajectory

different trajectories as a key predictor of subsequent reactivation during SWRs.

**Reactivated hippocampal-cortical patterns remain coherent.** Our results demonstrate that, in a familiar environment and task, SWR reactivation of hippocampal path location representations engages a subset of PFC cells expressing path-related representations that generalize across trajectories (i.e., cells with high all-trial similarity). These findings suggest a many-to-one mapping between hippocampal and PFC representations that is maintained in hippocampal-cortical networks, where many location representations in the hippocampus can be linked to a single generalized representation related to path traversal in the PFC. We tested this prediction by creating groups of SWRs that reactivated different hippocampal location representations, defining each group as the set of SWRs where only one place cell was reactivated. We then compared, across pairs of groups, the firing rate of concurrently reactivated PFC cells, focusing on PFC cells that expressed generalized representations ($R_{median} > 0.5$). As predicted, we found SWRs reactivating hippocampal place cells that represented locations on different trajectories could engage the same PFC cell (Fig. 6a).

Importantly, this many-to-one mapping reflected task relevant associations, and was not a result of non-specific activation of PFC cells across all SWRs. Here we took advantage of our recent demonstration that the reactivation of hippocampal representations for locations associated with movement (i.e., paths) is largely distinct from those for locations of immobility (i.e., wells)[33]. Based on that observation, we reasoned that generalized PFC path representations should be reactivated together with hippocampal path location representations but not with hippocampal well location representations.

Consistent with this prediction, we found PFC activity during SWRs was often different between SWRs defined by the reactivation of either a path or a well location active CA1 cell (Fig. 6b). Comparisons of PFC firing rates during these groups of

SWRs showed more significant differences than for comparisons between groups of path versus path place cell SWRs (see Methods section) (Fig. 6c). The PFC firing differences cannot be attributed to PFC modulation in response to differences in SWR power between path and well location reactivations (Supplementary Fig. 14). This indicates that PFC reactivation patterns distinguished between path versus well location reactivations, whereas different hippocampal path location reactivations were often associated with similar PFC reactivation patterns. These results confirm that the many-to-one mapping between hippocampal and cortical activity patterns reflected task information.

## Discussion
Our results identify a novel form of association between hippocampal and cortical representations that is reinstated during memory reactivation. In our well-trained animals, we found coordinated hippocampal and PFC activity during memory reactivation only weakly recapitulate their associations observed during ongoing experience, consistent with previous findings[32]. Nonetheless, a subset of PFC cells remains modulated during memory reactivation, and we found that these reactivated PFC cells preferentially represented general task features that were repeated across different trajectories rather than spatially specific elements of experience restricted to one or a few trajectories.

Our current results, together with recent findings[30,32] provide important insights on the transformation of memory representations with learning. We suggest the enrichment of many-to-one associations between hippocampal-cortical representations reflects the network's to ability to form representations that link frequently repeating features of ongoing experience (i.e., the repeating structure of each trial across multiple trajectories) simultaneously with features specific to an experience (i.e., traversal of one particular trajectory). During initial learning, stored hippocampal-cortical associations are likely to capture relationships specific to individual experiences, such as specific features of experience on each trajectory. At this stage, representations of

individual experiences are likely to remain unique since there are insufficient exposures for the network to recognize similarities between individual experiences. Through learning, links that map common features shared across experiences with specific features of single experiences become enriched whereas links between representations for specific features in the hippocampus and cortex diminishes. This would explain previous findings of weak correlations between ongoing and reactivated activity patterns after learning[32].

Learning-related transformations in memory representations parallel changes to representations observed during ongoing experience in the hippocampus and PFC. The hippocampus is likely to maintain enduring representations of specific spatial features of experience. In our task, hippocampal place cell representations showed highly place specific firing even after learning. In PFC, it was found that ensemble patterns for common task phases in two different contexts become more similar with experience, which suggests the encoding of general features

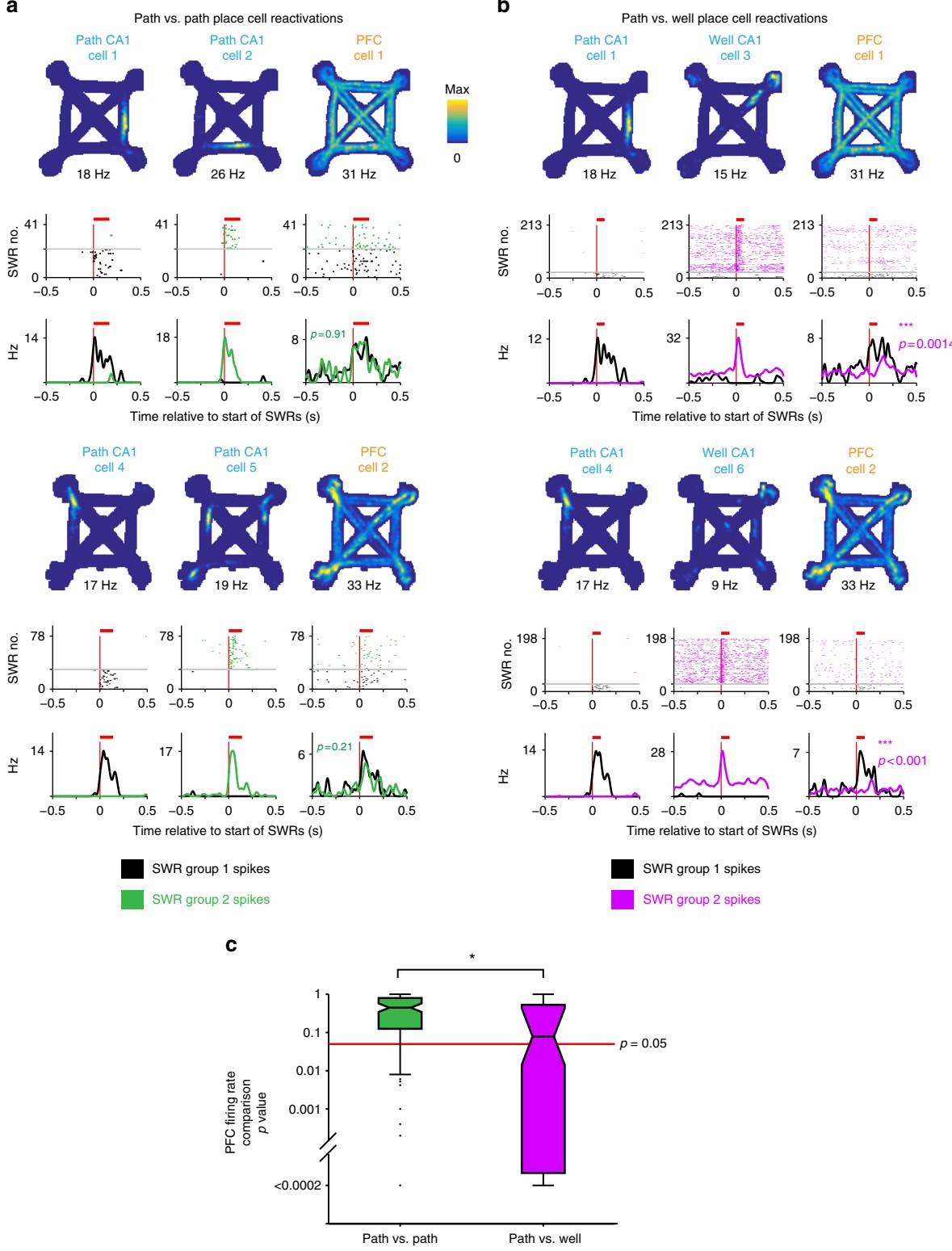

**Fig. 6** Similar PFC modulation accompanies distinct hippocampal path location reactivations. **a–b** Spatial firing rate map, SWR aligned spiking raster and firing rate for a pair of CA1 cells and a PFC cell that were recorded simultaneously. Each group of SWRs was defined as events containing only spikes from one of the two CA1 cells (columns 1 and 2). The corresponding spiking and firing rate of the PFC cell during these two groups of SWRs are shown in column 3. The mean duration of SWRs is indicated with a red bar. The difference in PFC firing rates between the two groups of SWRs is quantified using a permutation test. Two such sets are shown (CA1 cells 1–3 and PFC cell 1, and CA1 cells 4–6 and PFC cell 2). In the top set, CA1 cells 1 and 2, and PFC cell 1 corresponds to the CA1 cells and the PFC cell 4 in Fig. 1 respectively. PFC cell 2 in the bottom set corresponds to PFC cell a in Fig. 2. **a** Comparison of PFC spiking between SWRs containing different path place cell reactivations. **b** Comparison of PFC spiking between SWRs containing path or well place cell reactivations. **c** Boxplot of permutation test $p$ values for comparisons of PFC firing rates during SWRs between groups of SWRs. PFC cells with similar firing patterns across trajectories ($R_{median} > 0.5$) were included. PFC activity during SWRs reactivating different CA1 path location representations was similar (green, higher $p$ values, $N = 127$ SWR group pairs, examples in **a**) compared with activity during SWRs reactivating CA1 path or well location representations was dissimilar (magenta, lower $p$ values, $N = 87$ pairs, examples in **b**). Wilcoxon rank-sum test: ***$p < 10^{-3}$

of experience develop with learning[29]. While the mechanisms that drive changes in hippocampal-cortical associations across learning remain unknown, coordinated hippocampal-cortical communication during sleep may play a role[31,40–46]. The content of hippocampal-cortical memory reactivation during sleep SWRs is less similar to awake experience than reactivation during awake SWRs[32,47]. This raises the possibility that memory reactivation during sleep could reflect the reinstatement of representations for recent experiences that are interwoven with reinstated representations of existing knowledge. These "noisy" reactivation events could be involved in establishing more generalized cortical representations and in driving changes in hippocampal-cortical associations[48].

Enriching general-to-specific links in hippocampal-cortical networks could provide a basis for creating abstractions based on similar experiences as an animal learns[4–6], which requires the function of hippocampal-cortical networks[1–4,7,8,10,35,49–56]. This process could facilitate the creation of "schemas" or generalizable knowledge of a task[57–59]. The link between representations for general and specific representations of experience could serve to embed memories of experiences in a general knowledge structure while preserving the integrity of individual experiences. This in turn could enhance the brain's ability to use memories to guide future decisions[11,12].

## Methods

**Animal and behavior.** All experiments were conducted in accordance with University of California San Francisco Institutional Animal Care and Use Committee and US National Institutes of Health guidelines. No statistical methods were used to determine sample size. Experimenters were not blinded and no randomization was used. We trained 6 Long-Evans rats (male, 500–700 g, 4–9 months old) initially to traverse a 1 m long linear track for reward (evaporated milk plus 5% sucrose, Carnation). We then introduced the animals to the foraging task ~21 days after surgery. In the task, only two of the four possible reward well locations were chosen to deliver reward. The rat had to find those two well locations and visit them in alternation to receive reward. We changed the rewarded well locations within or between sessions, or between days[33]. These changes in spatial reward contingencies were not explicitly signaled and the rat needed to find the new rewarded well locations by trial and error. Data from 11, 9, 10, 12, 13 and 4 days were analyzed for each of the 6 animals respectively. Each day consisted of between 2–3 task sessions of 15–45 min each. The task sessions were interleaved with rest sessions of 20-60 minutes in a sleep box located away from the track. We used a custom built automated system for reward delivery, which was triggered by an infrared beam break at the well location. Reward was delivered immediately (two animals) or after a 1 s delay (four animals) after the beam break. A syringe pump (NE-500 OEM, New Era Pump Systems Inc.) delivered the reward (100–300 µl at 20 ml/min).

**Implant.** Custom designed and 3D printed (PolyJetHD Blue, Stratasys Ltd.) recording drives housed a maximum of 28 individually movable tetrodes. Tetrodes (Ni-Cr, California Fine Wire Company) were gold plated to reach an impedance of 250 kΩ at 1 kHz.

Implanted recording drives targeted both dorsal CA1 (7 tetrodes) and dorsal PFC (14–21 tetrodes, housed in one cannula angled at 20 degrees toward the midline). CA1 AP: −3.8 mm and ML: 2.2 mm. PFC (anterior cingulate cortex and dorsal prelimbic cortex): AP: + 2.2 mm, ML + 1.5 mm and DV between 1.88 mm to 2.72 mm depending on the AP and ML coordinates of each tetrode.

Tetrodes were adjusted every 2 days post-surgery to reach the target DV coordinate (PFC) or guided by LFP and spiking patterns (CA1). After the start of data acquisition, tetrodes were adjusted at the end of the day in small increments (typically ~30 µm) to improve cell isolation.

**Histology.** At the end of the experiment, we marked the location of recording sites by passing current through each tetrode (30 µA, 3 s) to create electrolytic lesions. After 12–24 h, we perfused the animals with paraformaldehyde (4% in PBS), fixed (24 h at room temperature) and cryoprotected the brain (30% sucrose in PBS at 4 °C). We identified the sites of electrolytic lesions with Cresyl Violet stained coronal sections (50 µm).

**Recording.** The NSpike data acquisition system (LMF and J. MacArthur, Harvard Instrumentation Design Laboratory) was used for data collection. Experiments were conducted in dim lighting. To track the animal's position, an infrared LED array was mounted on the headstage amplifier and video was recorded at 30 Hz. We recorded LFP from each tetrode (0.5–400 Hz sampled at 1.5 kHz). We recoded spiking activity from each recording channel (600–6000 or 300–6000 Hz sampled at 30 kHz). For hippocampal tetrodes, the reference for LFP and spike detection was a tetrode located in corpus callosum. For PFC, the reference was a tetrode located locally but did not detect spikes.

**Data preprocessing.** We identified putative neurons by manual clustering of spiking data from channels of each tetrode based on peak amplitude, spike width and wave-form principal components (MatClust, M.P.K.). Only stable and well-isolated cells were used for further analysis.

The animal's position was determined as the centroid of the front and back diodes from the LED array using a semiautomated analysis of the video.

**Cell selection.** CA1: We recorded from 391 CA1 neurons from which we excluded from our analysis putative fast spiking interneurons (spike peak to trough width <0.4 ms and mean firing rate >10 Hz, $N = 22$) and cells with < 200 spikes across all sessions on a given day ($N = 135$).

PFC: We recorded from 844 PFC from which we excluded from our analysis putative fast spiking interneurons (spike peak to trough width <0.3 ms and mean firing rate >7 Hz, $N = 42$) and cells that had <200 spikes across all sessions of a day ($N = 224$). We defined path active PFC cells as those with peak firing during the path phases of the trial.

**SWR detection.** The raw CA1 LFP was referenced to an electrode in corpus callosum and then filtered (150–250 Hz) to isolate the SWR band. The SWR envelope was then obtained using the Hilbert transform and convolved with a Gaussian kernel ($\sigma = 4$ ms). A consensus SWR envelope was calculated by taking the median of the envelopes across all available tetrodes. Only days where at least three tetrodes were in or near the CA1 cell layer were used for the analysis.

To avoid the problem of arbitrary thresholds and differences in noise distributions across days, we developed a SWR identification approach that defines a threshold based on the distribution of the consensus envelope power for a given day[33]. Only SWR events occurring at speed <4 cm/s and at reward well locations were included in our analyses. We chose these SWRs to separate in time, measures of trajectory-related PFC activity, which were measured during periods of movement between wells, and reactivation, which was measured during SWRs at the wells.

**Occupancy normalized firing maps.** The environment was first divided into 2 cm square bins. The occupancy-normalized rate was calculated by dividing the number of spikes by the occupancy of the animal per bin and smoothing with a two-dimensional symmetric Gaussian kernel ($\sigma = 2$ cm and 12 cm spatial extent).

**Trial normalization.** Each trial was defined as the time between entry to a well location and entry to the next well location. For each trial, the time the rat spent at

the well location was divided into 18 equally spaced time bins. The same was done to time when the rat was traveling between well locations. Thus, each trial comprises of 36 bins (Fig. 1b and Fig. 3a–b). Firing rate for each bin was calculated by dividing the number of spikes that occurred during each time bin by the duration of each time bin. The firing rate was smoothed using a Gaussian kernel ($\sigma = 1$ trial bin). Time and spiking during SWRs and within $\pm 50$ ms were excluded and does not contribute toward the calculation of firing properties (e.g., $R_{median}$ and $R_{max}$). The speed for each bin is the mean speed of time points in each bin. The speed profile for each trial was smoothed using a Gaussian kernel ($\sigma = 1$ trial bin).

**Trial firing and speed profile similarity.** All-trial firing similarity for each cell is the median of Pearson's correlation of firing profile between all trials ($R_{median}$, Fig. 1d). The same procedure was used to calculate trial speed profile similarity. To calculate maximum within-trajectory similarity, we first computed the median of pairwise Pearson's correlation between pairs of trials on a trajectory and repeated this for all trajectories. We then selected the maximum out of these values ($R_{max}$, Fig. 1e).

To ensure the animal's movement pattern was stereotyped across trials, both during times when the rat was at the reward location and on the path, we only included rewarded trials where the animal's overall probability of correct alternation was >0.75, which was estimated using a state-space model[60]. In addition, to ensure the reliability of the firing pattern of a cell on each trajectory was adequately sampled, only trajectories on which the animal made five or more traversals were included in the analyses. All-trial similarity and maximum within-trajectory reliability were only reported for cells with spiking during trials that fulfilled these selection criteria.

**Trial firing activity coverage.** Trial coverage was computed by first identifying the trial phase bin with the maximum firing rate. The number of consecutive bins surrounding the peak bin that exceeds 2/3 of maximum firing rate was then expressed as a proportion of the total number of trial phase bins.

**Speed-firing rate$_{residual\ correlation}$.** We first collected firing rate and speed values belonging to a task phase bin from all trials on the same trajectory, and then subtracted their means respectively. This process eliminated the baseline associated with each task phase, which allowed us to compute the correlation between firing rate and speed (speed-firing rate$_{residual\ correlation}$) for all data points independent of task phase. Real correlations between movement speed and firing rate will result in high residual correlation values, as trial phase bins with higher speed will have higher rates. This analysis controls for apparent correlations between firing rate and speed due to stereotyped movement patterns on each trial.

**PFC SWR modulation index.** The significance of modulation was calculated as describe previously[30,33,61]. We first generated a perievent time histogram (PETH) for all events aligned to the start of SWRs for the observed data. We then generated a control dataset by circularly permuting the spike times for each SWR event, such that all spikes around one SWR event were circularly shifted by the same amount but this amount varied between SWR events. From this control dataset, we then generated a PETH. This was repeated 1000 times. Next, we calculated the squared deviation of the observed PETH from the mean of the 1000 control PETHs for the average duration of SWRs for the given type of SWR. We then compared the squared deviation of each of the 1000 control PETHs to the mean of all 1000 control PETHs. The significance value was the fraction of 1000 control PETH deviations that are larger than the observed PETH deviation. We defined SWR reactivated path active PFC cells as those with a significant excitation during SWRs containing CA1 path place cells.

**Permutation test for distribution comparisons.** To normalize the observed differences in the $R_{median}$ (Fig. 5d) and $R_{max}$ (Fig. 5e) distributions between SWR reactivated and non-reactivated PFC populations, we used a permutation test to generate expected distributions of differences. First, we permuted the identities of SWR reactivated and non-reactivated PFC cells and generated their corresponding probability density functions (PDFs). We then calculated the difference between the PDFs of the two groups of the permuted dataset. We calculated the mean and standard deviation for each bin of the PDF using 10,000 permuted datasets with which we used to convert the observed data into z-scores.

**Resampling method for matching peri-SWR firing rates.** We controlled for the possibility that differences in peri-SWR firing rate could influence our ability to detect SWR modulation and give rise to the observed differences in all-trial similarity ($R_{median}$) between reactivated versus non-reactivated groups. This was achieved by resampling SWR reactivated and non-reactivated PFC cells to match their peri-SWR firing rates (Wilcoxon rank-sum test $p$ values > 0.05) (Supplementary Fig. 8A). The peri-SWR firing rate is the mean firing rate in a 1 s window centered on the start of SWRs. For each resampled dataset, we then obtained the $p$ value of the Wilcoxon rank-sum test for the corresponding all-trial similarity comparison (Supplementary Fig. 8B). This was repeated 1000 times. The proportion of the 1000 resamples with $R_{median}$ $p < 0.05$ was tested against the expected

proportion of resamples with $p < 0.05$ (5%) using a Binomial test (Supplementary Fig. 8C–D).

**PFC modulation between different SWR groups.** We compared the mean firing rate of two groups of unique SWRs, each containing a different hippocampal place cell. The place cells were active on paths (peak rate > 3 Hz) or active at a reward well location[33] (mean rate > 3 Hz). We extended this to all available pairs of place cells. The mean rate of a PFC cell during each group of SWRs was calculated by dividing the number of spikes observed in each SWR by the mean duration of all SWRs in both groups of a pairwise comparison. A permutation test (5000 permutations) was used to determine if the mean rate of the PFC cell is significantly different between the two groups of SWRs.

**Data availability.** Publicly available at the Collaborative Research in Computational Neuroscience data sharing website (http://crcns.org/; https://doi.org/10.6080/K0H41PK8)

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

## Acknowledgements

We thank B. Mensch, G. Rothschild, Mari Sosa, E. Anderson, D. Kastner, E. Phillips, C. Geaghan-Breiner, A. Gillespie, H. Joo and A. Kiseleva for comments on the manuscript. This work was supported by a Jane Coffin Childs Memorial Fund for Biomedical Research postdoctoral fellowship (J.Y.Y.), the Howard Hughes Medical Institute, NIH RO1MH105174, NIH R01MH097084 and University of California Office of the President Lab Fees Award #LF-12-237680 (L.M.F.).

## Author contributions

J.Y.Y and L.M.F. designed the experiments and wrote the manuscript. J.Y.Y. analyzed the data and performed experiments with assistance from I.G., D.F.L., and A.L.

## Additional information

**Competing interests:** The authors declare no competing interests.

