## [Peer Review File · Nature Communications]

Reviewers' comments:

Reviewer #1 (Remarks to the Author):

The authors have adequately responded to all of my previous comments. In particular, the new analyses and the addition of Figure 3 of the revised manuscript adds important information regarding the PFC representation. I remain enthusiastic about the findings and have no further concerns.

Reviewer #3 (Remarks to the Author):

In this study, the authors examine the task-coding properties of those neurons in the rat PFC that are active during sharp wave-ripple (SWR, "replay") events in the hippocampus. As in the original submission, the main finding is that during SWRs, a specific rather than randomly sampled group of PFC neurons is preferentially active: those neurons that tended to generalize over paths during experience (defined as high all-to-all trial similarity). Furthermore, CA1 neurons with trial-specific firing fields can be co-active with these generalizing PFC neurons. The authors interpret these results as linking general and specific features of experience in memory.

As before, I think the question of what information in PFC is co-active with hippocampal SWRs can provide an important window into the content and dynamics of memory. However, my reservations about the original submission have not been adequately addressed. As a result, I am still unconvinced that the results reflect anything novel or important. Specifically:

1) The authors need to provide more convincing control analyses to show that factors other than all-to-all trial similarity can explain the increased SWR activation probability in PFC. The authors include a GLM analysis in the current version, but as presented, it is unclear if this analysis is performed in the manner required to rule out the possibility that PFC cell firing rate or animal location.

In particular, the crucial question the GLMs should address is whether AFTER the effects of variables that reflect simpler explanations for the results -- such as mean and peak firing rate -- are accounted for, the addition of all-trial similarity improves the model (and by how much). Because the description of the GLM makes no mention of using such a sequential procedure, I assume the authors jointly fit all variables in the GLM, which does not address how much additional variance is explained by the key variable of interest.

This approach is required because if such an analysis is not performed, the alternative of "High-firing PFC neurons are preferentially reactivated during SWR events" could be an equally good as well as simpler account for the main result.

2) It needs to be unambiguously clear which SWR events are selected for the analyses reported, and if any such selection is important for the reported results.

To expand:

- It appears the main analysis (Figure 5) is performed on a subset of all SWRs. I understand that some selection needs to occur to make sure that least one PFC cell is available for analysis. However, in addition, it seems that there is an additional selection to include only those events during which at least one(?) path-preferring CA1 cell is active. Is this latter selection required to obtain the main result, or does the same outcome occur when SWRs with any CA1 cell, or all SWRs of equivalent power, are analyzed? In any case, the methods should contain a section explaining what different event selection steps are used for which analyses, and whether the results depend on them.

- Similarly, are SWRs included from when the animal was physically located on the track? If so, this could bias SWR content, and should be controlled for or ruled out by only analyzing off-track SWRs.

The above are important steps to document, because depending on the outcome, the main results may be better described as a simpler claim than is currently used, such as "task-related PFC cells are preferentially active during SWR events" regardless of what may be happening in CA1, or "PFC reactivation is biased by animal location" which I would not regard as particularly novel or interesting.

I expect the authors will be able to address the above technical points in a revision. That leaves the issue of what the significance of the preferential activation of "generalizing" PFC neurons is. In particular:

3) Ruling out the null hypothesis that all PFC neurons are equally likely to be active during SWRs is useful. However, the authors should also point out what would be expected if all paths are replayed equally during SWRs. In such a case, as I pointed out in my original review, any cell associated with multiple paths (such as the high all-all trial similarity cells) will be active during more path replays than cells that are associated with only one path. This prediction seems completely consistent with the authors' result, and neither requires nor suggests any notion of generalization beyond what is already known from the cells' firing fields during behavior (so, I would not regard it as "a novel form of association", line 201).

Including co-activation during behavior as a predictor of SWR activation, as the authors do in the rebuttal, does not seem to inform this issue because of the sequential nature of replay: a CA1 cell with a field at the start of a path and a PFC cell with a field at the end of a path may be co-active during sequential replay of that path, but depending on the binning used in the cross-correlation, need not be active in the same time bin during behavior. Similarly, I don't see how pointing out the Tang et al. result of a relatively weak relationship between behavior activity and SWR activity in familiar sessions strengthens their interpretation of the results. If anything, it seems to me this observation weakens the authors' case that SWR activity can be interpreted based on activity patterns during behavior! Thus, I think a disconnect remains between the results reported and the authors' interpretation of a novel form of association relevant to questions of generalization in memory.

In this study, the authors examine the task-coding properties of those neurons in the rat PFC that are active during sharp wave-ripple (SWR, "replay") events in the hippocampus. As in the original submission, the main finding is that during SWRs, a specific rather than randomly sampled group of PFC neurons is preferentially active: those neurons that tended to generalize over paths during experience (defined as high all-to-all trial similarity). Furthermore, CA1 neurons with trial-specific firing fields can be co-active with these generalizing PFC neurons. The authors interpret these results as linking general and specific features of experience in memory.

As before, I think the question of what information in PFC is co-active with hippocampal SWRs can provide an important window into the content and dynamics of memory. However, my reservations about the original submission have not been adequately addressed. As a result, I am still unconvinced that the results reflect anything novel or important. Specifically:

1) The authors need to provide more convincing control analyses to show that factors other than all-to-all trial similarity can explain the increased SWR activation probability in PFC. The authors include a GLM analysis in the current version, but as presented, it is unclear if this analysis is performed in the manner required to rule out the possibility that PFC cell firing rate or animal location.

In particular, the crucial question the GLMs should address is whether AFTER the effects of variables that reflect simpler explanations for the results -- such as mean and peak firing rate -- are accounted for, the addition of all-trial similarity improves the model (and by how much). Because the description of the GLM makes no mention of using such a sequential procedure, I assume the authors jointly fit all variables in the GLM, which does not address how much additional variance is explained by the key variable of interest.

This approach is required because if such an analysis is not performed, the alternative of "High-firing PFC neurons are preferentially reactivated during SWR events" could be an equally good as well as simpler account for the main result.

We addressed Reviewer 3's concern by performing GLMs sequentially and asking how much the explained variance increases with the addition of more predictors. We note first that we carried out a similar analysis for the previous round of reviews where we showed that none of these other variables were significant predictors of SWR engagement when trial similarity measures were included in the model. We nevertheless carried out the analysis exactly as the reviewer described to verify the minimal contributions of these variables.

We found mean firing rate, trial phase coverage and a measure of speed-firing correlation based on residual firing rates (Speed-firing rate_{residual correlation}), could only explain a small percentage of the variance of the prediction (Figure S12, blue, green and orange bars). This is true for including one, a combination of two or all three measures in the GLM. When we included trial firing similarity parameters to the GLM, the explained variance was greatly increased (Figures S12, red bars). These sequential GLMs indicate the

major contributor to predicting SWR engagement is trial similarity rather than various combinations of firing rate, trial phase coverage or speed correlation.

Figure S12. Trial similarity measures contribute more than trial firing measures when predicting SWR engagement.

Explained variance was calculated for GLMs with a combination of variables to predict SWR engagement for PFC cells. The inclusion of trial similarity measures greatly increased the explained variance of the prediction.

2) It needs to be unambiguously clear which SWR events are selected for the analyses reported, and if any such selection is important for the reported results.

To expand:

- It appears the main analysis (Figure 5) is performed on a subset of all SWRs. I understand that some selection needs to occur to make sure that least one PFC cell is available for analysis. However, in addition, it seems that there is an additional selection to include only those events during which at least one(?) path-preferring CA1 cell is active. Is this latter selection required to obtain the main result, or does the same outcome occur when SWRs with any CA1 cell, or all SWRs of equivalent power, are analyzed? In any case, the methods should contain a section explaining what different event selection steps are used for which analyses, and whether the results depend on them.

We agree that it is important to specify the exact set of SWRs that were included, and we have added text to the Results and Methods to make this clear (marked in red).

In addition, as per Reviewer 3's suggestion, we repeated our analysis with all SWRs, not just the ones with identified place cells active on the track. The results remain consistent (Fig. S11). Thus, our findings are not due to the selection of subsets of SWR events.

While we now mention this result in the text, there are good reasons not to include all SWRs in these analyses. We showed previously that there are distinct types of SWRs that engage either movement-associated or immobility-associated hippocampal place cells (Yu et al., 2017). These SWRs in turn engage different sets of PFC cells, and thus using all SWRs combines these two types. In this case the results turn out the same way, likely because the majority of SWRs engage movement-related representations (Yu et. al. 2017). Nonetheless, as the content does matter, there are strong scientific reasons to focus on SWRs where we can determine that the hippocampal content involves reactivations of path-related information.

Figure S11. Enrichment of PFC cells with high R_{median} remains when all SWRs were included in calculating SWR modulation.

A. Distribution of R_{median} distributions for SWR modulated (orange) and not modulated (gray) PFC cells. Kolmogorov–Smirnov test: $***p < 10^{-3}$.

B. Difference in the R_{median} distributions between path SWR reactivated and not reactivated PFC cells normalized using a permutation test.

- Similarly, are SWRs included from when the animal was physically located on the track? If so, this could bias SWR content, and should be controlled for or ruled out by only analyzing off-track SWRs.

We have now emphasized in the revised text how we selected the SWRs. SWRs occur during immobility, predominantly at the well locations in the task. As we now make clear in the Methods, we only included events at the well locations. Since we are analyzing path-related reactivation content, the content represents locations away from the physical location of the animal where the SWRs occur.

The above are important steps to document, because depending on the outcome, the main results may be better described as a simpler claim than is currently used, such as "task-related PFC cells are preferentially active during SWR events" regardless of what may be happening in CA1, or "PFC reactivation is biased by animal location" which I would not regard as particularly novel or interesting.

We agree with Reviewer 3 that these simpler explanations would not be interesting, but our analysis addressed all of these concerns. The SWRs we examined occur when the animal is at well locations and not on the paths connecting the well locations. We examined

“path” active PFC content during these SWRs, which corresponds to times/locations not matching the animal’s current location. This eliminates bias from the animal’s current location to the content of reactivation that we examined.

I expect the authors will be able to address the above technical points in a revision. That leaves the issue of what the significance of the preferential activation of “generalizing” PFC neurons is. In particular:

3) Ruling out the null hypothesis that all PFC neurons are equally likely to be active during SWRs is useful. However, the authors should also point out what would be expected if all paths are replayed equally during SWRs. In such a case, as I pointed out in my original review, any cell associated with multiple paths (such as the high all-all trial similarity cells) will be active during more path replays than cells that are associated with only one path. This prediction seems completely consistent with the authors’ result, and neither requires nor suggests any notion of generalization beyond what is already known from the cells’ firing fields during behavior (so, I would not regard it as “a novel form of association”, line 201).

Including co-activation during behavior as a predictor of SWR activation, as the authors do in the rebuttal, does not seem to inform this use because of the sequential nature of replay: a CA1 cell with a field at the start of a path and a PFC cell with a field at the end of a path may be co-active during sequential replay of that path, but depending on the binning used in the cross-correlation, need not be active in the same time bin during behavior.

To summarize, Reviewer 3 is concerned that a PFC cell active on one path is likely to participate in a subset of SWRs reactivating that path, whereas a PFC cell active on two paths is likely to participate in another subset of SWRs, albeit a larger one. Reviewer 3 is in effect arguing that our SWR modulation detection method is not sensitive enough to capture modulation that occurs in only a fraction of events, resulting in a bias against identifying PFC cells that participate in fewer events.

We note that individual CA1 neurons only participate in a small fraction of SWRs, but our method reliably detects CA1 modulation during SWRs, as demonstrated in our recent publication¹.

In any case, the central prediction of the reviewer’s scenario is that a PFC cell that fires on one trajectory during behavior will be active during SWRs where that trajectory is reactivated. If so, CA1-PFC co-activity on a trajectory will predict CA1-PFC coactivity during SWRs (as it does early during learning). To further demonstrate that this is not the case in more familiar settings, we repeated our co-activation analysis across a 5 second window, computing the maximum cross-correlation within this period. The length of the time window was chosen because it approximates the travel time for individual trajectories².

The results of this analysis show that these longer timescale behavioral correlations are actually weakly *negatively* correlated with SWR reactivation (Fig. R1). This result, together with the analysis in the initial rebuttal (Fig. S13A), indicates ongoing coactivity,

even in a larger time window, does not predict SWR coactivity in the way Reviewer 3 envisioned.

Figure R1. Maximum cross-correlation within a 5s window is weakly correlated with SWR correlation. Hippocampal-PFC cell pairs that were coactivity during ongoing experience but were not reactivated together are circles in red. We computed coactivity between CA1-PFC cell pairs, during ongoing experience, we used the same subset of path-active PFC cells and calculated their spike cross-correlation over a 5 second window for times when the rat is on the path. We took the maximum z-scored values the 5 second window as the measure of peak cross-correlation. A high cross-correlation value indicates coactivity between CA1-PFC cell pairs during ongoing experience. To compute coactivity during SWRs, we then calculated the correlation the spike counts of the same pairs of CA1-PFC cells in a 200ms aligned to the start of SWRs. The correlation value is normalized by subtracting the mean and dividing by the standard deviation of the correlation values for 5000 shuffles where the identities of the reactivation events are permuted. A high correlation z indicates a CA1-PFC cell pair has coordinated firing during SWRs.

Similarly, I don't see how pointing out the Tang et al. result of a relatively weak relationship between behavior activity and SWR activity in familiar sessions strengthens their interpretation of the results. If anything, it seems to me this observation weakens the authors' case that SWR activity can be interpreted based on activity patterns during behavior!

Here we think there is confusion about the results of the Tang et. al. paper. The Tang paper shows that spatial coactivity, measured as the tendency for cells to fire together during a short time window during behavior, is predictive of SWR reactivation in novel

environments. This demonstrates that early during learning when a situation is novel SWR activity can be interpreted based on activity patterns during behavior.

The Tang et. al. results also show that once the task and the environment are familiar, this simple relationship between spatiotemporal co-activity and SWR co-activity is no longer present. Given that many PFC cells still show modulation during SWRs, this raised the important question of whether there is a different, non-spatial overlap-based rule that explains the SWR activity of PFC cells. Our results provide such a rule. We provide evidence that the interpretation of coordinated content during SWR activity needs to look beyond the coordination of spatial representations between hippocampus and PFC. Our work highlights the importance of considering select subsets of non-spatial but behaviorally relevant coding schemes in PFC that are linked to spatial representations in hippocampus. Our current work provides a deeper insight to our understanding of coordinated hippocampal-cortical activity that has not been described before.

- 1 Rothschild, G., Eban, E. & Frank, L. M. A cortical–hippocampal–cortical loop of information processing during memory consolidation. *Nature Neuroscience* **20**, 251-259, doi:10.1038/nn.4457 (2016).
- 2 Yu, J. Y. *et al.* Distinct hippocampal-cortical memory representations for experiences associated with movement versus immobility. *eLife* **6**, doi:10.7554/eLife.27621 (2017).

Reviewers' comments:

Reviewer #3 (Remarks to the Author):

As expected, the authors satisfactorily addressed my technical comments with additional analyses in this revision. Most importantly of these additions is the demonstration that the best-fitting model of PFC cell participation in SWRs includes inter-trajectory similarity.

On the one hand, this analysis provides rigorous support for the authors' main claim that the degree to which PFC cells generalize across trajectories is related to SWR participation.

On the other hand, I think it is noteworthy that this analysis reveals that the addition of this factor - the basis of the motivation and key claims in the paper - improves the model fit by a modest ~2.5% of total variance. In the interest of transparency, I hope the authors will consider including an explicit statement about this effect size in the main text, and not just in the crucial Figure S12.

My other main comment concerned the interpretation of the results: can the observed pattern be explained by already known properties of SWR content? One example of such an explanation I raised in my previous comments: assuming coordinated replay between HC and PFC, a PFC cell with high inter-trajectory similarity would be expected to have increased SWR participation by virtue of co-occurring with multiple trajectories during behavior. The authors provide evidence that this scenario is unlikely in Figure R1, because HC-PFC co-occurrence during behavior is in fact anticorrelated with SWR co-activation. However, this suggests something of a paradox: SWR content apparently avoids linking trajectory-specific cells and the PFC cells they actually co-occurred with during behavior (as indicated by the negative correlation in Figure R1), but instead links with *other* (non co-occurring) PFC cells. Is such active exclusion of specific trajectories consistent with generalization? The authors should comment on this issue.

A different alternative explanation for the observed results has been made more urgent by new information provided in this revision, that the SWRs analyzed occurred primarily at the reward wells on the task. This introduces an obvious potential confound because the animal's location biases hippocampal SWR content (Davidson et al. 2009; Carr et al. 2011). If, like HC cells, PFC cells with firing fields close to the animal's location are more likely to participate in SWRs, then increased participation follows trivially from having multiple firing fields (as implied by high inter-trajectory similarity). To exclude this interpretation, the authors should determine if spatial proximity is related to SWR participation in PFC cells.

If this can be done, I would be satisfied that the authors have appropriately dealt with alternative explanations.

Reviewer #3 (Remarks to the Author):

As expected, the authors satisfactorily addressed my technical comments with additional analyses in this revision. Most importantly of these additions is the demonstration that the best-fitting model of PFC cell participation in SWRs includes inter-trajectory similarity.

On the one hand, this analysis provides rigorous support for the authors' main claim that the degree to which PFC cells generalize across trajectories is related to SWR participation.

On the other hand, I think it is noteworthy that this analysis reveals that the addition of this factor - the basis of the motivation and key claims in the paper - improves the model fit by a modest ~2.5% of total variance. In the interest of transparency, I hope the authors will consider including an explicit statement about this effect size in the main text, and not just in the crucial Figure S12.

We have added an explicit statement about the effect size in the main text. We note the improvement in explained variance is 4.4% after the inclusion of Intratrajectory R_{median} but up to 7.8% for Intertrajectory R_{median} in that version of Fig. 12.

My other main comment concerned the interpretation of the results: can the observed pattern be explained by already known properties of SWR content? One example of such an explanation I raised in my previous comments: assuming coordinated replay between HC and PFC, a PFC cell with high inter-trajectory similarity would be expected to have increased SWR participation by virtue of co-occurring with multiple trajectories during behavior. The authors provide evidence that this scenario is unlikely in Figure R1, because HC-PFC co-occurrence during behavior is in fact anticorrelated with SWR co-activation. However, this suggests something of a paradox: SWR content apparently avoids linking trajectory-specific cells and the PFC cells they actually co-occurred with during behavior (as indicated by the negative correlation in Figure R1), but instead links with *other* (non co-occurring) PFC cells. Is such active exclusion of specific trajectories consistent with generalization? The authors should comment on this issue.

Figure R1 showed a significant but weak negative correlation between maximum cross-correlation during a 5 second window and SWR reactivation for PFC cell pairs. This result cannot be used to make the strong conclusion that *“SWR content apparently avoids linking trajectory-specific cells and the PFC cells they actually co-occurred with during behavior (as indicated by the negative correlation in Figure R1), but instead links with *other* (non co-occurring) PFC cells.”*

The actual values of cross-correlations were still high, $\sim 2z$ for SWR-coactivated cell pairs that are coactivated during SWRs. This is inconsistent with the reviewer 3's interpretation that completely uncorrelated CA-PFC pairs are coactivated together during SWRs. The negative correlation comes from the majority of CA-PFC cell pairs not being coactivated during SWRs. In fact, only ~30-40% of PFC cells show significant SWR modulation even if they are coactive with CA1 cells during ongoing experience, which has been demonstrated previously¹⁻³. A valid conclusion based on these results is that with

familiarity, CA1-PFC coactivity during ongoing behavior is a poor predictor of SWR coactivity, which is consistent with previous findings³.

Thus, that result does not create a paradox. It reinforces the notion that ongoing coactivity cannot fully explain CA1-PFC coordination during SWR reactivation. Our results identify an organizing principle that can explain the pattern of coordination between the two regions.

A different alternative explanation for the observed results has been made more urgent by new information provided in this revision, that the SWRs analyzed occurred primarily at the reward wells on the task. This introduces an obvious potential confound because the animal's location biases hippocampal SWR content (Davidson et al. 2009; Carr et al. 2011). If, like HC cells, PFC cells with firing fields close to the animal's location are more likely to participate in SWRs, then increased participation follows trivially from having multiple firing fields (as implied by high inter-trajectory similarity). To exclude this interpretation, the authors should determine if spatial proximity is related to SWR participation in PFC cells.

If this can be done, I would be satisfied that the authors have appropriately dealt with alternative explanations.

We found spatial proximity does not explain our observations using sequential GLMs (Fig. R1), a method that we used to rule out other potential explanatory factors per the referee's suggestion. We computed spatial proximity, expressed as the number of trial bins from firing rate peak bin to the nearest well bin. This was done since our maze has multiple wells. We found proximity to wells can only explain <2% of the total variance of the prediction. With the addition of trial similarity measures, we can increase explained variance by ~8% (~4 fold improvement) for Intertrajectory R_{median} . We have added this additional control to Fig. S12.

Contribution of PFC firing proximity to wells
and trial similarity measures
on GLM prediction

Fig. R1. Trial similarity measures contribute more than PFC firing proximity to reward well for SWR participation prediction.

Figure S12. Trial similarity measures contribute more than trial firing measures for predicting SWR modulation.

Explained variance was calculated for GLMs with a combination of variables to predict SWR modulation. The inclusion of trial similarity measures greatly increased the prediction.

References

- 1 Yu, J. Y. *et al.* Distinct hippocampal-cortical memory representations for experiences associated with movement versus immobility. *eLife* **6**, doi:10.7554/eLife.27621 (2017).
- 2 Jadhav, S. P., Rothschild, G., Roumis, D. K. & Frank, L. M. Coordinated Excitation and Inhibition of Prefrontal Ensembles during Awake Hippocampal Sharp-Wave Ripple Events. *Neuron* **90**, 113-127, doi:10.1016/j.neuron.2016.02.010 (2016).
- 3 Tang, W., Shin, J. D., Frank, L. M. & Jadhav, S. P. Hippocampal-prefrontal reactivation during learning is stronger in awake as compared to sleep states. *The Journal of neuroscience : the official journal of the Society for Neuroscience*, doi:10.1523/JNEUROSCI.2291-17.2017 (2017).

REVIEWERS' COMMENTS:

Reviewer #3 (Remarks to the Author):

The addition of proximity between the animal's location and the peak of the PFC cell firing field to the GLM is important, because it rules out the possibility that the known bias to include the animal's current location in CA1 cell reactivation accounts for the observed results.

Looking at Figure S12 however, it seems that the best-fitting GLM that includes the crucial variable of interest -- inter-trajectory R_{median} -- explains about 10.5% of variance, and the best fitting model that does not include that variable (but does include intra-trajectory R_{max}) explains about 8.5%. Thus, the variance uniquely attributable to inter-trajectory R_{median} , rather than other variables not related to the generalization concept of interest, is about 2%.

Regarding the point about how CA1-PFC correlations during behavior relate to SWR co-activation: one way or another, the authors need to come up with a clear and consistent explanation of what is novel about these results. As I have pointed out previously, known properties of CA1 and PFC reactivation, when put together, seem to imply exactly what the authors find. That does not mean the results are uninteresting, but in order to claim "a novel form of association" the authors really do need to rule out the following account (expanded here from a previous comment for additional precision):

Premise 1: any trajectory, such as a path between two wells on the task, has a certain probability of being replayed by CA1 cells during a given SWR.

Example 1: The path between wells W1 and W2 is associated with the sequential activation of CA1 cells A, B, and C. The paths between wells W2 and W3 is associated with the sequential activation of CA1 cells D, E, and F. Some SWRs show co-activation of drawn from cells A, B, and C; other SWRs show co-activation drawn from cells D, E, and F.

Corollary 1: the set of SWRs that involve co-activation drawn from ABC is smaller than the set of SWRs that involve co-activation drawn from ABC or DEF.

Premise 2: SWR activity is coordinated between CA1 and PFC.

Example 2: A PFC cell that is active between wells W1 and W2 is more likely to reactivate during a SWR when the corresponding CA1 cells (ABC) reactivate, than during a SWR when non-corresponding CA1 cells (DEF) reactivate.

Conclusion: A PFC cell that is active BOTH between wells W1 and W2, and between wells W2 and W3, is reactivated during more SWRs than a PFC cell that is active between wells W1 and W2 only. (In other words, the main finding.)

Based on the authors' most recent reply, it is still unclear if any of the above premises are wrong, if they are correct but inadequate to explain the observed effect, or something else.

Minor comments:

Line 230: typo, "...reflects the network's to ability to..."

Figure S13 legend has items A-D but figure has panels A-E

Reviewer #3 (Remarks to the Author):

The addition of proximity between the animal's location and the peak of the PFC cell firing field to the GLM is important, because it rules out the possibility that the known bias to include the animal's current location in CA1 cell reactivation accounts for the observed results.

Looking at Figure S12 however, it seems that the best-fitting GLM that includes the crucial variable of interest -- inter-trajectory R_{median} -- explains about 10.5% of variance, and the best fitting model that does not include that variable (but does include intra-trajectory R_{max}) explains about 8.5%. Thus, the variance uniquely attributable to inter-trajectory R_{median} , rather than other variables not related to the generalization concept of interest, is about 2%.

While it is tempting to interpret these results in this way, it is not mathematically correct. Since R_{median} and intra-trajectory R_{max} are correlated, one cannot measure the difference between the explained variance in the two cases and interpret it as due to one variable or another. In this case we think it is more accurate to say that R_{median} and intratrajectory R_{max} capture related but not identical aspects of generalization without making specific claims about their relative importance. Our results are consistent with these firing similarity measures contributing to prediction of participation in SWR reactivation more than other firing properties of the cell.

Regarding the point about how CA1-PFC correlations during behavior relate to SWR co-activation: one way or another, the authors need to come up with a clear and consistent explanation of what is novel about these results. As I have pointed out previously, known properties of CA1 and PFC reactivation, when put together, seem to imply exactly what the authors find. That does not mean the results are uninteresting, but in order to claim "a novel form of association" the authors really do need to rule out the following account (expanded here from a previous comment for additional precision):

Premise 1: any trajectory, such as a path between two wells on the task, has a certain probability of being replayed by CA1 cells during a given SWR.

Example 1: The path between wells W1 and W2 is associated with the sequential activation of CA1 cells A, B, and C. The paths between wells W2 and W3 is associated with the sequential activation of CA1 cells D, E, and F. Some SWRs show co-activation of drawn from cells A, B, and C; other SWRs show co-activation drawn from cells D, E, and F.

Corollary 1: the set of SWRs that involve co-activation drawn from ABC is smaller than the set of SWRs that involve co-activation drawn from ABC or DEF.

Premise 2: SWR activity is coordinated between CA1 and PFC.

Example 2: A PFC cell that is active between wells W1 and W2 is more likely to reactivate during a SWR when the corresponding CA1 cells (ABC) reactivate, than during a SWR when

non-corresponding CA1 cells (DEF) reactivate.

Conclusion: A PFC cell that is active BOTH between wells W1 and W2, and between wells W2 and W3, is reactivated during more SWRs than a PFC cell that is active between wells W1 and W2 only. (In other words, the main finding.)

Based on the authors' most recent reply, it is still unclear if any of the above premises are wrong, if they are correct but inadequate to explain the observed effect, or something else.

This argument is correct, but incomplete. The key additional element, which captures the novelty of our result, is that a PFC cells that is only active between wells W1 and W2 will typically not be reactivated during SWRs, DESPITE the coactivity with the hippocampal cells active on that trajectory. This stands in contrast to what happens in a novel environment, where that PFC cell would tend to be reactivated during SWRs.

Thus, our result is novel because it identifies something other than simple environmental co-activity that predicts SWR reactivation in PFC.

Minor comments:

Line 230: typo, "...reflects the network's to ability to..."

This is not a typo.

Figure S13 legend has items A-D but figure has panels A-E

We have corrected this error in the text.